# A Low Rate of Periprosthetic Infections after Aseptic Knee Prosthesis Revision Using Dual-Antibiotic-Impregnated Bone Cement

**DOI:** 10.3390/antibiotics12091368

**Published:** 2023-08-25

**Authors:** Benedikt Paul Blersch, Michael Barthels, Philipp Schuster, Bernd Fink

**Affiliations:** 1Department of Joint Replacement, General and Rheumatic Orthopaedics, Orthopaedic Clinic Markgröningen gGmbH, Kurt-Lindemann-Weg 10, 71706 Markgröningen, Germany; benedikt.blersch@rkh-gesundheit.de (B.P.B.); michael.barthels@rkh-gesundheit.de (M.B.); philipp.schuster@rkh-gesundheit.de (P.S.); 2Department of Orthopaedics and Traumatology, Paracelsus Medical University, Prof. Ernst Nathan Straße 1, 90419 Nuremberg, Germany; 3Orthopaedic Department, University Hospital Hamburg-Eppendorf, Martinistrasse 52, 20246 Hamburg, Germany

**Keywords:** knee revision arthroplasty, antibiotic-impregnated bone cement, periprosthetic joint infection

## Abstract

Aim: The incidence of periprosthetic joint infections (PJI) following aseptic knee revision arthroplasty lies between 3% and 7.5%. The aim of this study was to verify the hypothesis that the use of dual-antibiotic-impregnated cement in knee revision arthroplasty leads to a lower rate of periprosthetic joint infections. Methods: We retrospectively reviewed 403 aseptic revision knee arthroplasties performed between January 2013 and March 2021 (148 revisions of a unicompartmental prosthesis, 188 revisions of a bicondylar total knee arthroplasty (TKA), 41 revisions of an axis-guided prosthesis, and 26 revisions of only one component of a surface replacement prosthesis). The bone cement Copal G+C (Heraeus Medical, Wertheim, Germany) with two antibiotics—gentamycin and clindamycin—was used for the fixation of the new implant. The follow-up period was 53.4 ± 27.9 (4.0–115.0) months. Results: Five patients suffered from PJI within follow-up (1.2%). The revision rate for any reason was 8.7%. Survival for any reason was significantly different between the types of revision (*p* = 0.026, Log-Rank-test), with lower survival rates after more complex surgical procedures. The 5-year survival rate with regard to revision for any reason was 91.3% [88.2–94.4%] and with regard to revision for PJI 98.2% [98.7–99.9%], respectively. Conclusion: The use of the dual-antibiotic-impregnated bone cement Copal G+C results in a lower rate of periprosthetic infections after aseptic knee prosthesis replacement than that reported in published prosthesis revisions using only one antibiotic in the bone cement.

## 1. Introduction

Periprosthetic joint infection (PJI) is a relatively rare but serious complication of primary knee arthroplasties, with an incidence of approximately 0.5–2% [1,2,3,4]. In aseptic knee revision arthroplasty, the infection rate is much higher, ranging from 3% to 7.5% [5,6]. Therefore, reducing the risk and rate of periprosthetic joint infection in revision total knee arthroplasty is an important goal.

PJIs are most frequently caused by Gram-positive bacteria, primarily by *Staphylococcus aureus* and by *Staphylococcus epidermidis* [7,8]. It is estimated that Gram-negative bacteria only account for 12–15% of PJIs [9,10]. Overall *Pseudomonas aeruginosa*, *Escherichia coli,* and *Klebsiella pneumoniae* are the most commonly detected pathogens in monomicrobial Gram-negative PJIs [9,10]. The treatment of Gram-negative PJIs is considered to be more challenging due to increasing antibiotic resistance of Gram-negative bacteria [10,11].

Antibiotic loaded bone cement (ALBC) for fixation of the prosthetic implant is used in order to complement the systemic antibiotic prophylaxis to reduce the risk of PJI. ALBC delivers antibiotics straight into the vulnerable compartment of the joint. This leads to local antibiotic concentrations higher than the minimum inhibitory antibiotic concentration of potential bacteria. The avoidance of high systemic antibiotic concentrations potentially causing side effects is beneficial [12]. Furthermore, the combination of systemic antibiotic prophylaxis and ALBC gains is important because of the increasing prevalence of resistant bacteria. The resistance to the most commonly used systemic perioperative antibiotics, cefazolin and cefuroxime, limits the efficacy of systemic antibiotic prophylaxis [13,14].

The use of two antibiotics in the bone cement for the fixation of the new implants leads to a synergistic effect, and the elution of each individual antibiotic is better in dual-impregnated cement than the individual antibiotic alone in single-impregnated cement [15,16,17,18,19]. Therefore, bone cement containing gentamycin and clindamycin in combination (Copal G+C, Heraeus Medical GmbH, Wertheim, Germany) is used in our institution instead of bone cement containing gentamycin alone (Palacos R+G) in aseptic knee revision arthroplasty. Gentamycin exhibits a concentration-dependent bactericidal effect: It is effective against Gram-positive bacteria (*Staphylococcus aureus*, coagulase-negative *Staphylococci*, *Enterococcus species*), as well as some Gram-negative bacteria (*Klebsiella*, *Escherichia coli*, *Acinetobacter*, *Pseudomonas*, *Proteus*, *Citrobacter*, *Enterobacter*, *Serratia*) and mycobacteria. Clindamycin shows a time-dependent bacteriostatic effect in low dosages and is effective against Gram-positive bacteria *(Staphylococci* and *Streptococci species*) as well as anaerob pathogens (*Cutibacteria*, *Peptostreptococcus*, *Bacteroides*). However, Cara et al. could show that the dual ALBC Copal G+C seems to be the most effective ALBC for preventing PJI with Gram-negative bacteria [20].

There is limited data in the literature regarding the reduction in periprosthetic infections after aseptic knee revision arthroplasty using dual ALBC. In the only reported study, Sanz-Ruiz et al., on the basis of 246 patients (143 cases with dual ALBC Copal G+C and 103 cases with single low dose ALBC Palacos R+G) analyzed retrospectively, found no cases of PJI in the Copal G+C group compared to 6 cases occurring in the Palacos R+G group (PJI rate = 4.1%, *p* = 0.035) [21].

Therefore, the aim of the current study was to verify the hypothesis that the use of the dual ALBC Copal G+C in knee revision arthroplasty leads to a low rate of periprosthetic joint infections on the basis of a high number of cases for the first time.

## 2. Results

Thirty-two patients (7.9%) underwent further revision surgery during the follow-up period, of which five patients (1.24%) were revised because of periprosthetic joint infection (PJI) (Table 1). In one patient, debridement, irrigation, and implant retention (DAIR) were performed 4 weeks postoperative for an acute early infection, and four patients underwent one-stage (three patients) or two-stage septic prosthesis replacement (one patient) because of a delayed infection.

Kaplan–Meier analysis showed a 5-year survival rate ranging from 82.5% to 100% for revision for any reason, and from 97.2% to 100% for septic revision (Table 2, Figure 1, Figure 2, Figure 3 and Figure 4). Survival for revision for any reason was significantly different between the types of revision (*p* = 0.026, Log-Rank-test) with lower survival rates in more complex cases with constrained implants surgeries. Survival for periprosthetic infection (with only five cases throughout the study) was not different between the groups (*p* = 0.927, Log-Rank-test). With regard to survival, there were no statistically significant differences with regard to ASA classification (*p* = 0.944) or CCI (*p* = 0.163), respectively.

## 3. Discussion

The infection rate of 1.24% overall (0.25% of acute periprosthetic infections, and 1% of delayed periprosthetic infections (>4 weeks after the implantation)) is lower than the rates of 3–7.5% overall reported in the literature [5]. The demographic data of the patients in our study with dual-impregnated bone cement did not differ from that described in the previous reports of the literature using single-impregnated bone cements with higher rates of PJI [15,16,17,18,19,20]. Therefore, it can be assumed that the use of the dual-impregnated bone cement Copal G+C in our study had a positive effect on the risk of suffering from periprosthetic infection after aseptic revision of knee arthroplasties.

The positive effect of cement containing the two antibiotics gentamycin (G) and clindamycin (C) for the prophylaxis of PJI in patients at higher risk as well as in the treatment of PJI is attributable to several factors:1.With regard to the spectrum of activity, the two antibiotics act synergistically and thus broaden their range of effectiveness. This means that virtually all pathogens relevant to periprosthetic infections (PJI) are targeted by the two antibiotics [20,22].2.With respect to the elution of the individual antibiotics, both G and C are eluted more efficiently in the combination in dual ALBC than when alone in the cement. The molecules of both antibiotics are small and hydrophilic and have very good diffusion properties [22].3.Due to the higher local release of the antibiotics, the otherwise bacteriostatic antibiotic clindamycin reaches local concentrations that make it bactericidal [22].4.The mode of action or target of the antibiotics in bacteria is also synergistic in that gentamycin targets the 30 s ribosome on the mRNA while clindamycin has a different site of action, namely the 50 s ribosome. This means that the bacteria are attacked at two different sites at the same time, providing a synergy in the prevention of bacterial resistance development [22].

Ensing et al. were able to show in vitro that the dual-antibiotic-impregnated cement Copal G+C cement was more effective in preventing bacterial biofilm formation than the single-impregnated cement Palacos R+G [17].

In other instances, beside aseptic knee revision arthroplasty with a high risk of periprosthetic joint infection (PJI) or surgical side infection (SSI), several studies could show a reduced rate of PJI and SSI when the dual-antibiotic-loaded bone cement (ALBC) Copal G+C was used [23,24,25]. Sprowson et al. analyzed 848 patient cases of intracapsular femoral neck fractures (FNF) treated with hip hemiarthroplasty using either dual ALBC Copal G+C (intervention group, n = 400) or single low dose ALBC Palacos R+G (control group, n = 448) [23]. They identified a significantly lower incidence of deep surgical site infections (SSI) in the intervention group with the applied dual ALBC Copal G+C when compared to the control group with the single low dose ALBC Palacos R+G. In the intervention group the deep SSI rate was 1.1% and in the control group 3.5% (*p* = 0.041). Tyas et al. expanded in a quasi-randomized controlled trial the number of patient cases (n = 1941) and compared the incidence of deep SSI using dual ALBC Copal G+C (n = 1260) and of deep SSI using single low dose ALBC Palacos R+G (n = 681) in hip hemiarthroplasty of FNF, as well [24]. The authors verified the lower rate of PJI of the dual ALBC group and presented an incidence of deep SSI using dual ALBC Copal G+C of 1.2% versus an incidence of deep SSI using single low dose ALBC Palacos R+G of 3.4%.

Savage et al. analyzed retrospectively 206 cases of cemented hip arthroplasty, including hemiarthroplasty as well as total hip replacements, for patients with intracapsular FNF [25]. They retrospectively compared 102 patients undergoing cemented hip arthroplasty for femoral neck fracture with single-antibiotic-impregnated cement to 98 prospectively studied patients with the same surgery but utilizing dual-antibiotic-impregnated cement. They found a rate of periprosthetic infections of 2.9% in the first group, whereas no periprosthetic infections (0%) occurred in the second group [25].

To the best of our knowledge Sanz-Ruiz et al. published the only study testing the effect of dual ALBC Copal G+C on the PJI rate in revision total knee arthroplasty [21]. To test the hypothesis of dual ALBC Copal G+C being a potent prophylaxis of PJI in aseptic revision knee arthroplasty, Sanz-Ruiz et al. reviewed 246 patient cases retrospectively. Thereof, in 143 cases dual ALBC Copal G+C and in 103 cases single low dose ALBC Palacos R+G was used for implant fixation. Septic and oncological causes of revision were excluded. They observed no cases of PJI in the Copal G+C group compared to six cases occurring in the Palacos R+G group (PJI rate = 4.1%, *p* = 0.035). These results led them to conclude that more effective PJI prevention is achieved by using dual ALBC in aseptic revision knee arthroplasty. This observation is confirmed by the outcomes of the present study involving a significantly higher number of aseptic revision surgeries of knee prostheses.

Comparing the different types of replacement surgeries in our study, there was no difference in the survival for periprosthetic infection between the groups. However, for the survival for revision for any reason a significantly higher risk of revision surgery was seen for rotating hinged prostheses that had been changed to a rotating hinged prosthesis. Axis-guided prostheses generally have higher revision rates, according to the observations of Houfani et al. [26]. In a multicenter study of 127 rotating-hinge prostheses implanted during revision surgery, they found a 5-year survival rate of 77% (95% confidence interval, 0.70–0.85) after a mean observation time of 67.3 ± 11.8 months (range, 13–180 months). Postoperative complications developed in 29% of patients (infection, n = 12; aseptic loosening, n = 11; and fracture, n = 7). The National Joint Register of England, Wales, and Northern Ireland reported a 5-year revision rate for hinged prostheses of 10.7% (95% CI 9.6 to 11.9) [27]. This is consistent with the revision rate in our study for a rotating hinge prosthesis (Table 1).

This study has strengths and weaknesses. One strength is the homogeneous approach to a very large number of cases. To the best of our knowledge, this is the study with the largest case number of aseptic knee prosthesis changes with Copal G+C cement. However, the case number of the present study is not large enough to statistically identify risk factors for revisions and septic revisions in particular. The number of septic revisions was too small for statistical evaluation of possible risk factors using multiple regression analysis. Future multicenter studies with larger numbers of cases are needed in this regard. It must also be noted as a weakness that there is no control group with the use of a cement with only one antibiotic. Here, only data from published studies in the literature could be utilized. The latter shows that the infection rate when using Copal G+C cement is lower than that found in the literature for single-impregnated bone cements [5]. The minimum follow-up of 2 years is sufficient for the study of postoperative periprosthetic infections. Regarding the temporal and causal relationship between periprosthetic infections (PJIs) and the prosthesis implantation that took place, PJIs in the first 4 to 12 weeks are classified as acute postoperative infections and PJIs up to 2 years postoperatively as delayed infections. The latter may still have a causal relationship with the initial operation [28,29,30]. Moreover, reinfections after one- or two-stage septic knee revision arthroplasties usually occur within the first two postoperative years [31,32].

## 4. Materials and Methods

All 403 aseptic knee prosthesis revisions performed from January 2013 to March 2021 and preoperative exclusion of a PJI were re-examined retrospectively. In all 403 patients, the new prosthesis was implanted with Copal G+C cement (Copal G+C, Heraeus Medical GmbH, Wertheim, Germany). A total of 42.7 g of this high viscosity, radiopaque bone cement contains 1.0 g gentamicin (as gentamicin sulphate) and 1.0 g clindamycin (as clindamycin hydrochloride).

The cohort consisted of 273 females and 130 males, aged 75.8 ± 9.9 (45.0–99.0) years (Figure 5). There were 120 revisions (29.8%) of a medial unicompartmental prosthesis to bicondylar total knee arthroplasty (TKA), 8 revisions (2.0%) of a lateral unicompartmental prosthesis to TKA, 20 revisions (5.0%) of a medial unicompartmental prosthesis to a rotating hinge prosthesis, 188 revisions (46.7%) from a TKA to a rotating-hinge prosthesis, 41 revisions (10.2%) from a rotating-hinge to a rotating-hinge prosthesis, and 26 (6.5%) revisions of one component of a TKA. BMI was 30.6 ± 5.8 kg/m^2^ (18.8–52.9). The time between primary surgery and aseptic revision surgery was 76.7 ± 61.0 (1.0–319.0) months. With respect to the American Society of Anesthesiologists (ASA) classification scores, 27 patients were classed as ASA 1, 220 patients were ASA 2, 152 patients ASA 3, and 4 patients ASA 4 [33,34]. Regarding the Charlson Comorbidity Index (CCI), there were 190 patients with CCI 0, 100 patients with CCI 1, 51 patients with CCI 2, 32 patients with CCI 3, 16 patients with CCI 4, 8 patients with CCI 5, 3 patients with CCI 6, and 3 patients with CCI 7 [34,35].

Regarding secondary diseases potentially relevant to the development of PJI, 3 patients (0.7%) had type I diabetes mellitus, 82 patients (20.3%) had type II diabetes mellitus, 22 patients (5.5%) had rheumatoid arthritis, 1 patient (0.2%) had HIV, 98 patients (24.3%) hypothyroidism, and 294 patients (73.0%) cardiovascular disease (hypertension, myocardial infarction, atrial fibrillation, congestive heart failure, valvular vitium, pulmonary circulation disorder, peripheral arterial disease, thromboembolism, or chronic venous insufficiency). Thirty-seven patients (9.2%) had been treated with immunosuppressive agents (corticosteroids, disease-modifying antirheumatic drugs). PJI of another joint was in the medical records of 10 patients (2.5%). Perioperative allogeneic blood transfusion was given to 16 patients (4.0%).

All patients had undergone arthrocentesis of the corresponding knee joint to test for periprosthetic joint infection (PJI) prior to revision surgery. PJI was excluded by determining the cell count and alpha-defensin level as well as culturing the aspirate for 14 days additional to the preoperative serum CPR level using the ICM criteria [36,37]. Moreover, to rule out PJI intraoperatively five samples of the synovial tissue and periprosthetic tissue were taken for culture analysis and five additional samples for histological examination (Figure 5).

Hereby, the synovia samples were immediately sent to the microbiology unit. Under sterile conditions the samples were minced and homogenized, and under standard conditions they were incubated on different agars: chocolate agar, Columbia agar, McConkey agar, and Schaedler agar. The incubation was performed for 2–3 days under aerobic conditions and for 5 days under anaerobic conditions.

In addition, the samples were incubated in thioglycolate broth and in Brain Heart Infusion (BHI) for 14 days, as previously described by Schäfer et al. [36], Steinbrink et al. [38], Atkins et al. [39], and Virolainen et al. [40]. Bacterial growth in thioglycolate broth and BHI was checked daily. In agreement with Schäfer et al. [36], Steinbrink and Frommelt [38] and Ince et al. [41] clear growth media despite 14 days of incubation were defined as negative. Turbid media were subcultured onto appropriate agar plates (s.a.). On all samples, obligatory staining for Gram negative bacteria was performed.

To detect samples with remaining antibiotics that may lead to false negative results, the incubation of one aliquot on Bacillus subtilis agar was conducted in order to assess the presence of antimicrobial activity.

In case antimicrobial activity was detected, the bacterial colonies were picked, and MALDI TOF MS (Biomerieux Vitek MS, Nürtingen, Germany) was used for strain identification. In special cases, the strain identification was, in addition, supported using the Analytical Profile Index System (Biomerieux, Nürtingen, Germany).

The results of culture analyses and histological examination were analyzed in accordance with the criteria of the Musculoskeletal Infection Society (MSIS) [42] and the ICM-2018-Definition [37], as well as according to Atkins et al. [39], Pandey et al. [43], and Virolainen et al. [40]. If at least one of the two following conditions below was fulfilled, the sample of the synovial membrane was considered positive:1.Identification of at least two synovial membrane samples with bacterial growth showing the same pathogen.2.Identification of at least one synovial membrane sample with bacterial growth in combination with a histological examination showing at least five neutrophilic polymorph leukocytes in five high power fields (×400) and an increased CPR-value (>10 mg/L), as described in the MSIS-criteria [42] and in the ICM-2018-Definition [36], as well as in accordance with Feldman et al. [44].

According to Virolainen et al., bacterial growth in a singular sample without a histological correlation of PJI was assumed to be a consequence of contamination occurring either during the sampling procedure or during the incubation period [40].

During the revision surgery, patients received antibiotic prophylaxis for 24 h in each case (3rd generation cephalosporin or vancomycin if allergic to the former).

Patients were contacted and questioned regarding any revision that had taken place and, specifically, revision due to periprosthetic infection. Follow-up time was 53.4 ± 27.9 (4.0–115.0) months. In total, 49 patients (12.2%) had a follow-up shorter than 24 months.

Data was obtained and analyzed retrospectively. Statistical analysis was performed using IBM SPSS Statistics (version 24, IBM Corp., Armonk, NY, USA). Cumulative survival was calculated using the Kaplan–Meier method with 95% confidence intervals (95%-CI); survival curves are shown as Kaplan–Meier plots. A log-rank test was used for comparison of survival rates. Unless otherwise stated, descriptive results are expressed as mean ± standard deviation (and range) or absolute number (percentage), respectively. The level of significance was set at *p* < 0.05.

## 5. Conclusions

The use of the dual-antibiotic-impregnated bone cement Copal G+C results in a very low rate of periprosthetic infections after aseptic knee arthroplasty revisions and should, therefore, in our opinion, be preferred to a cement with only one antibiotic in these revision surgeries to reduce the risk of periprosthetic joint infection (PJI). Further multicenter studies should be performed with high case numbers and control groups to evaluate the value of dual ALBC in revision arthroplasty surgeries as well as other surgeries with a higher risk of PJI in comparison to single-antibiotic-impregnated bone cements.

## Figures and Tables

**Figure 1 antibiotics-12-01368-f001:**
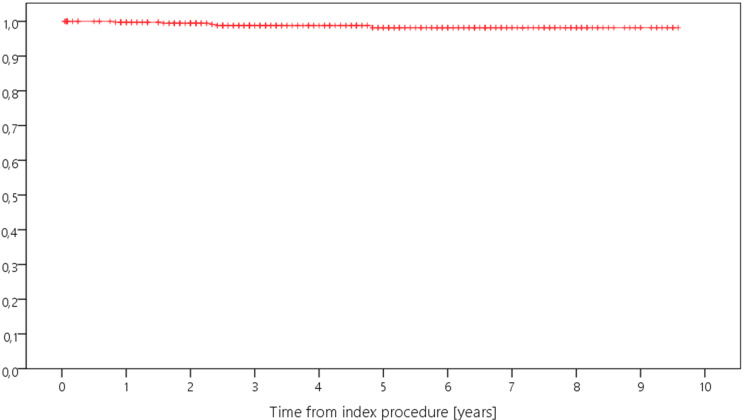
Kaplan–Meier curve displaying the estimated survival probability of the whole study group concerning revision for any reason (*x*-axis: years from index surgery; *y*-axis: percentage/100).

**Figure 2 antibiotics-12-01368-f002:**
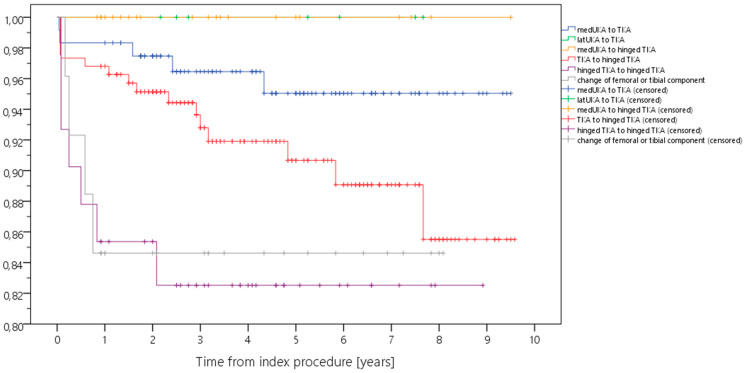
Kaplan–Meier curve displaying the estimated survival probability of the different groups concerning revision for any reason (*x*-axis: years from index surgery; *y*-axis: percentage/100).

**Figure 3 antibiotics-12-01368-f003:**
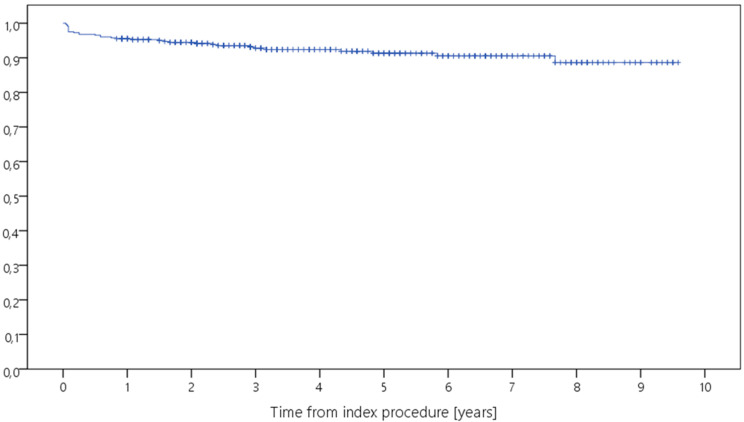
Kaplan–Meier curve displaying the estimated survival probability of the whole study group concerning revision for PPI (*x*-axis: years from index surgery; *y*-axis: percentage/100).

**Figure 4 antibiotics-12-01368-f004:**
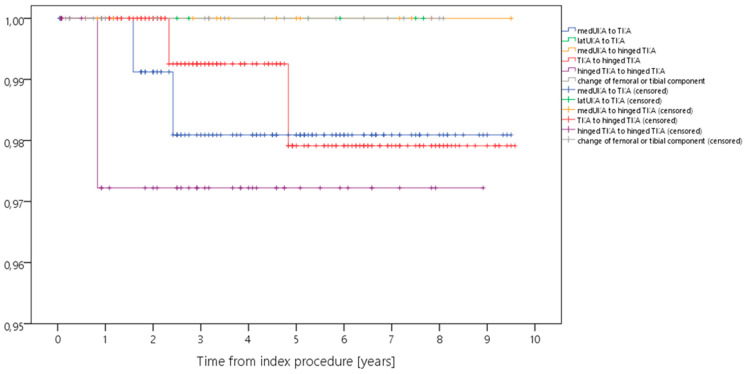
Kaplan–Meier curve displaying the estimated survival probability of the different groups concerning revision for PPI (*x*-axis: years from index surgery; *y*-axis: percentage/100).

**Figure 5 antibiotics-12-01368-f005:**
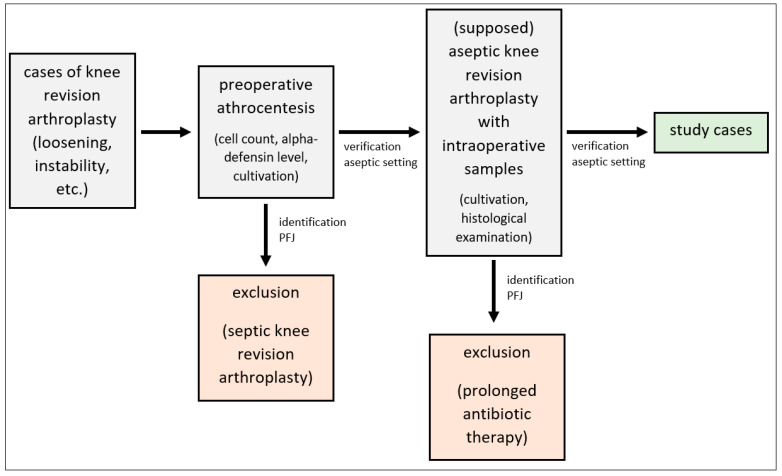
Flow chart demonstrating the selection of study cases.

**Table 1 antibiotics-12-01368-t001:** Reason for revision.

	N	Periprosthetic Joint Infection	Aseptic Loosening	Hematoma	Low Range of Motion	Dislocation of Insert	Periprosthetic Fracture	Superficial Wound Healing Problems without Infection	
**medial uni- to bicondylar TKA**	**120**	2		1	1	1			**5**
**lateral uni- to bicondylar TKA**	**8**		1						**1**
**medial uni- to rotating hinge**	**20**								**0**
**bicondylar to rotating hinge**	**188**	2	6		1	1	2	3	**15**
**rotating hinge to rotating hinge**	**41**	1	1				4	1	**7**
**exchange of only one component of bicondylar TKA**	**26**		1				2	1	**4**
		**5**	**9**	**1**	**2**	**2**	**8**	**5**	**32**

**Table 2 antibiotics-12-01368-t002:** Cumulative 5-Year-Survival Rates [with 95%-CI].

Survival Cohort	Survival for Revision for Any Reason	Survival for Revision for PJI
whole study group	91.3% [88.2–94.4%]	98.2% [98.7–99.9%]
medial UKA to TKA	95.0% [90.7–99.3%]	98.1% [95.5–100.0%]
lateral UKA to TKA	100.0%	100.0%
medial UKA to hinged TKA	100.0%	100.0%
TKA to hinged TKA	90.7% [85.8–95.6%]	97.9% [95.0–100.0%]
hinged TKA to hinged TKA	82.5% [70.7–94.3%]	97.2% [91.9–100.0%]
change of femoral or tibial component	84.6% [70.7–98.5%]	100.0%

## Data Availability

We do not wish to share our data, because some of patient’s data regarding individual privacy, and according to the policy of our hospital, the data could not be shared to others without permission.

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
