# Peer review of "A Low Rate of Periprosthetic Infections after Aseptic Knee Prosthesis Revision Using Dual-Antibiotic-Impregnated Bone Cement"

_antibiotics, 2023, doi:10.3390/antibiotics12091368_

Round 1

Reviewer 1 Report

The work seems interesting. However, there is a significant drawback concerning its presentation. The authors are strongly encouraged to resubmit their work after a substantial revision taking into account the following points:

1. The title of the manuscript is too big, it is advised to short it.

2. What edge does this paper have over the existing works? This needs to be presented as a novelty statement in the introduction.

3. Introduction is written simply; most recent research and innovation on this work must be added to align the work as latest research. There is not any stastics data as well to mention where these biomaterials have huge demand or not?

4. The novelties, contributions and objectives of the study should be addressed in the last paragraph of the Introduction. It is not clear what are the novelties of the study. 

5. In sentence no. 79, In figure 1, It is suggested to mention the  title of the y axis of the graph.

6. In sentence no. 83, In figure 2, It is suggested to mention the  title of the y axis of the graph.

7. In sentence no. 89, In figure 3, It is suggested to mention the  title of the y axis of the graph.

8. In sentence no. 93, In figure 4, It is suggested to mention the  title of the y axis of the graph.

9. The section 4 (Materials and method) should be as section 2, just after "Introduction section".

10. In sentence 256, In conclusion section, future of this work must be aligned in revised article.

11.Highlight the future scope of present research work.

12. Sentence no.99, In discussion section, it is difficult to ascertain the meaning or significance of the authors’ findings due to unclear writing. A more simplified narrative is required.

13.Pay close attention to style guidelines (formatting for references/citations in text and equation formatting).

Author Response

REVIEWER 1: The answers to the comments are in red.

The work seems interesting. However, there is a significant drawback concerning its presentation. The authors are strongly encouraged to resubmit their work after a substantial revision taking into account the following points:

  1. The title of the manuscript is too big, it is advised to short it. Is done
  2. What edge does this paper have over the existing works? This needs to be presented as a novelty statement in the introduction. Is mentioned at the end of the introduction.
  3. Introduction is written simply; most recent research and innovation on this work must be added to align the work as latest research. There is not any stastics data as well to mention where these biomaterials have huge demand or not? Is done
  4. The novelties, contributions and objectives of the study should be addressed in the last paragraph of the Introduction. It is not clear what are the novelties of the study. Is done
  5. In sentence no. 79, In figure 1, It is suggested to mention the  title of the y axis of the graph. Is done
  6. In sentence no. 83, In figure 2, It is suggested to mention the  title of the y axis of the graph. Is done
  7. In sentence no. 89, In figure 3, It is suggested to mention the  title of the y axis of the graph. Is done
  8. In sentence no. 93, In figure 4, It is suggested to mention the  title of the y axis of the graph. Is done
  9. The section 4 (Materials and method) should be as section 2, just after "Introduction section". The organization of the series of the sections is according to the guidelines of the journal. We are happy to change that if the editors want that.
  10. In sentence 256, In conclusion section, future of this work must be aligned in revised article. Is done at the end of the conclusion section.

11.Highlight the future scope of present research work. Is done at the end of the conclusion section.

  1. Sentence no.99, In discussion section, it is difficult to ascertain the meaning or significance of the authors’ findings due to unclear writing. A more simplified narrative is required. The discussion section is rewritten

13.Pay close attention to style guidelines (formatting for references/citations in text and equation formatting). Is done

Reviewer 2 Report

This study sets out with a clear aim: to investigate whether dual antibiotic-impregnated cement in knee revision arthroplasty reduces the incidence of PJI. The problem is clearly stated. The manuscript appears to be scientifically valid and technically sound in methodology. The study's methodology appears robust, with a sizable sample of 403 aseptic revision knee arthroplasties retrospectively reviewed over an eight-year period. The clear breakdown of types of revisions provides comprehensive coverage of the subject. The results are promising. With only a 1.2% incidence of PJI within the follow-up period, the study evidences a rate lower than the general aseptic knee revision statistics. The study also highlights a significant variation in survival rates depending on the type of revision, with more complex procedures exhibiting lower survival rates.

Specific feedback points:

1. Within the Introduction, there is an omission of background information on the antibiotics employed, specifically Gentamicin's efficacy against gram-negative bacteria and Clindamycin's action against anaerobic bacteria. This information is belatedly introduced in lines 145-148.

2. In the Methods section, I advise the incorporation of a flow chart delineating the study design to increase reader comprehension.

3. Regarding to the Results section, I recommend refining the visual presentation of the images to increase their appeal. Currently, the imagery is somewhat ambiguous, making it challenging to discern the presented data.

In conclusion, this paper provides compelling evidence for the efficacy of dual antibiotic-impregnated cement in reducing PJI incidence post-aseptic knee revisions. It would be interesting to see future research delve deeper into the reasons behind the overall 8.7% revision rate, exploring ways to further enhance post-surgical outcomes. It is salient to note that the authors have conscientiously acknowledged the study's limitations.

Author Response

Reviewer 2: The answers to the comments are in red

This study sets out with a clear aim: to investigate whether dual antibiotic-impregnated cement in knee revision arthroplasty reduces the incidence of PJI. The problem is clearly stated. The manuscript appears to be scientifically valid and technically sound in methodology. The study's methodology appears robust, with a sizable sample of 403 aseptic revision knee arthroplasties retrospectively reviewed over an eight-year period. The clear breakdown of types of revisions provides comprehensive coverage of the subject. The results are promising. With only a 1.2% incidence of PJI within the follow-up period, the study evidences a rate lower than the general aseptic knee revision statistics. The study also highlights a significant variation in survival rates depending on the type of revision, with more complex procedures exhibiting lower survival rates.

Specific feedback points:

  1. Within the Introduction, there is an omission of background information on the antibiotics employed, specifically Gentamicin's efficacy against gram-negative bacteria and Clindamycin's action against anaerobic bacteria. This information is belatedly introduced in lines 145-148. Is done. The introduction section is rewritten.
  2. In the Methods section, I advise the incorporation of a flow chart delineating the study design to increase reader comprehension. Is done as figure 5.
  3. Regarding to the Results section, I recommend refining the visual presentation of the images to increase their appeal. Currently, the imagery is somewhat ambiguous, making it challenging to discern the presented data. The visualization of the images is improved by the enlargement of the images.

In conclusion, this paper provides compelling evidence for the efficacy of dual antibiotic-impregnated cement in reducing PJI incidence post-aseptic knee revisions. It would be interesting to see future research delve deeper into the reasons behind the overall 8.7% revision rate, exploring ways to further enhance post-surgical outcomes. It is salient to note that the authors have conscientiously acknowledged the study's limitations.

Reviewer 3 Report

The reduce of postoperative complications is a general targhet in surgery and there were developed many strategies in this reson, one of it to prevent the infections is to use the antibiotics. The molecules of  gentamycin and clindamycin that are used in the study are small and hydrophilic showing a very good diffusion properties, but having different site of action on the bacteria. The authors  noted as a weakness that there is no control group with the use of a cement with only one antibiotic, but they compared their results with the same of literature. The number of patients, 403, their periodic control during 2 years and the collection of cases between 2013 and 2021 permitted to do an accurate work focused on the knee post-prosthetic healing process taking in account that some patients suffered by secondary diseases potentially relevant to the development of PJI.

Author Response

Reviewer 3: The answers to the comments are in red

The reduce of postoperative complications is a general targhet in surgery and there were developed many strategies in this reson, one of it to prevent the infections is to use the antibiotics. The molecules of  gentamycin and clindamycin that are used in the study are small and hydrophilic showing a very good diffusion properties, but having different site of action on the bacteria. The authors  noted as a weakness that there is no control group with the use of a cement with only one antibiotic, but they compared their results with the same of literature. The number of patients, 403, their periodic control during 2 years and the collection of cases between 2013 and 2021 permitted to do an accurate work focused on the knee post-prosthetic healing process taking in account that some patients suffered by secondary diseases potentially relevant to the development of PJI.

No changes are needed
